# Potential Application of TALENs against Murine Cytomegalovirus Latent Infections

**DOI:** 10.3390/v11050414

**Published:** 2019-05-03

**Authors:** Shiu-Jau Chen, Yuan-Chuan Chen

**Affiliations:** 1Department of Neurosurgery, Mackay Memorial Hospital, Taipei 10449, Taiwan; chenshiujau@gmail.com; 2Department of Medicine, Mackay Medicine College, New Taipei City 25245, Taiwan; 3Comparative Biochemistry program, University of California, Berkeley, CA 94720, USA; 4National Applied Research Laboratories, Taipei 10636, Taiwan

**Keywords:** cytomegalovirus, latent infection, TALEN, Surveyor nuclease mutation detection assay, *ie-1* gene, quantitative real-time PCR

## Abstract

Cytomegalovirus (CMV) infections are still a global health problem, because the latent viruses persist in humans and cause recurring diseases. Currently, there are no therapies for CMV latent infections and the therapies for active infections are limited by side effects and other problems. It is impossible to eradicate latent viruses in animals. HCMV (human CMV) is specific to human diseases; however, it is difficult to study HCMV due to its host specificity and long life cycle. Fortunately, MCMV (murine CMV) provides an excellent animal model. Here, three specific pairs of transcription activator-like effector nuclease (TALEN) plasmids (MCMV1–2, 3–4, and 5–6) were constructed to target the MCMV M80/80.5 sequence in order to test their efficacy in blocking MCMV lytic replication in NIH3T3 cell culture. The preliminary data showed that TALEN plasmids demonstrate specific targeting and cleavage in the MCMV M80/80.5 sequence and effectively inhibit MCMV growth in cell culture when the plasmid transfection is prior to the viral infection. The most specific pairs of TALEN plasmids (MCMV3–4) were further used to confirm the negative regulation of latent MCMV replication and gene expression in Balb/c mice. The injection of specific TALEN plasmids caused significant inhibition in the copy number level of immediately early gene (*ie-1*) DNA in five organs of mice, when compared with the controls. The result demonstrated that TALENs potentially provide an effective strategy to remove latent MCMV in animals.

## 1. Introduction

Studies on the functions of viral genes in human cytomegalovirus (HCMV) replication in vivo and understanding of viral pathogenesis are essential for developing novel drugs and strategies to treat the viral infections, but there are no suitable animal models for HCMV infection at present. Because HCMV proliferates in human cells specifically, grows slowly and has a very long lytic replication cycle in humans, it is quite difficult to study HCMV (genome size 235 kb) gene function and pathogenesis [1]. However, the murine cytomegalovirus (MCMV) provides an excellent animal model for studying the biology of cytomegaloviruses (CMV) through its specific infection of mice. An MCMV genome of 230 kb is predicted to encode more than 170 open reading frames (ORFs), 78 of which have an extensive homology to those of HCMV [1,2,3]. Moreover, the pathogenesis of MCMV infection in mice is very similar to that of HCMV infection in humans in several aspects such as active infections, establishment of latency, and reactivation after latent infections [1,3,4,5]. A complete understanding of the biology of MCMV and the function of its genes may provide insights into the pathogenesis of HCMV.

As a structural protein, the HCMV UL80 assembly protein is presumed to function in packaging DNA. A protease is encoded by the N-terminal region of UL80 that cleaves the assembly protein precursor at a site near the C terminus. The MCMV homolog (M80) of the assembly protein of HCMV (UL80) conserves the domain structure and cleavage sites present in HCMV UL80. MCMV is different in that the conserved CD1 and CD2 domains are separated by 100 amino acids, whereas in all other sequenced herpes viruses the two domains are separated by 80 to 84 residues [2]. The MCMV homolog (M80.5) of HCMV, UL80.5 ORF, is referred to code for protease. The ORFs of MCMV M80 and M80.5 are required for the assembly of proteins and proteases (capsid synthesis) and further virion production. The transcriptions of adjoining MCMV M80 and M80.5 ORFs have different start sites; however, they have the same stop site. Therefore, the overlapping region of MCMV M80 and M80.5 (M80/80.5) is an appropriate target site for virus inhibition.

Transcription activator-like effectors (TALEs) are proteins secreted by the plant pathogenic bacteria *Xanthomonas* via a type III secretion system where they infect various plant species [6]. These proteins can bind promoter sequences in the host plant and activate the expression of host genes that aid bacterial infections. TALEs are important virulence factors that act as transcriptional activators in the plant cell nucleus, where they directly bind to DNA via a central domain of tandem repeats [6]. Each TALE contains a central repetitive region consisting of varying numbers of repeat units (about 17.5 repeats) of 34 amino acids [6,7]. The DNA-binding domain contains a highly conserved 34-amino acid sequence, with the exception of the 12th and 13th amino acids [6]. Only the 12th and 13th amino acids in TALEs are changeable and variable; the other amino acids are constant and stable. These two locations’ RVDs (Repeat Variable Di-residues) are highly variable and show a strong correlation with a specific nucleotide recognition by different frequencies, for example NI recognizes A (55%), NG recognizes T (50%), NN recognizes G (7%) and HD recognizes C (69%) [6]. DNA transcribes RNA according to the complementary base pairing rule (A = U, G ≡ C) and RNA translates protein according to the standard genetic code (RNA codon table). After the breaking of the code of the DNA-binding specificity of TALEs [6,7], we know that two amino acids can also recognize one nucleotide. The restriction endonuclease *Fok* I, naturally found in the bacterium *Flavobacterium okeanokoites*, consists of an N-terminal specific DNA-binding domain and a C-terminal nonspecific DNA cleavage domain. The DNA-binding domain recognizes the non-palindromic sequence 5′-GGATG-3′ when the catalytic domain cleaves double-stranded DNA nonspecifically at a fixed distance of 9 and 13 nucleotides downstream of the recognition site. *Fok* I exists as an inactive monomer and becomes an active dimer upon binding to its target DNA and in the presence of specific divalent metals [7]. The DNA cleavage domain of *Fok* I functions as a homodimer, requiring two constructs with unique DNA-binding domains for sites in the target genome with proper orientation and spacing. Transcription activator-like effector nucleases (TALENs) are artificial restriction enzymes generated by fusing the specific TALE DNA-binding domain to a nonspecific *Fok* I DNA cleavage domain [8,9] (Figure 1).

TALENs were shown to be a valuable tool for precise genome engineering with low toxicity [10]. Additionally, they have been successfully applied in the genetic engineering of human pluripotent cells [11,12], and the generation of knockout animals, such as nematodes (*Caenorhabditis elegans*) [8], rats [13], and zebrafish [14,15]. TALENs can be used to edit genomes by making double-stranded breaks (DSBs), which cells respond to with two repair mechanisms: non-homologous end joining (NHEJ) or homologous recombination (HR). DNA can be introduced into a genome through NHEJ in the presence of exogenous double-stranded DNA fragments. HR can also introduce foreign DNA at the DSB as the transfected double-stranded sequences are used as templates for the repair enzymes. However, TALENs have some potential problems. If TALENs do not specifically target a unique site within the genome of interest, off-target cleavage may occur [11]. Such off-target cleavage may lead to the production of enough DSB to overcome the repair machinery and consequently result in chromosomal rearrangements and/or cell death [8].

Previous reports have suggested that TALENs can be engineered to adapt for an antiviral strategy. For example, TALENs were known to be effective in the inactivation of Hepatitis B virus (HBV) replication in cultured cells and in vivo [16], and in the targeting of the HBV genome [17]. Also, Epstein–Barr virus (EBV)-encoded nuclear antigen-1 (EBNA1) plays a crucial role in EBV episome replication and persistence. TALEN-mediated targeted disruption of EBNA1 was shown to inhibit the growth of EBV-infected cells, hinting at a possible therapeutic application for EBV-associated disorders [18]. TALENs may also provide a new strategy for the treatment of CMV infections.

Latency is a specific phase in viral life cycles, in which viral particles stop producing after infection, but the viral genome has not been completely removed. Proviral latency and episomal latency are two known viral latency models. CMV belongs to the episomal latency model which is essentially quiescent in myeloid progenitor cells, and is reactivated by differentiation, inflammation, immunosuppression or critical diseases [19]. CMV latency has been defined as the absence of infectious viruses, despite the presence of viral DNA. Although the molecular mechanisms by which latency is established and maintained have not been clear, transcriptional control of viral gene expression is very important in controlling viral latency and reactivation. Viral replication is initiated by the expression of *ie* (immediately early) genes. Studies with CMV have suggested that latency is established through the repression of *ie-1* gene expression. *Ie-1* proteins, the first proteins expressed by the virus during productive infection, are transcriptionally regulatory proteins that are required for the induction of early and late gene expression, viral DNA synthesis, and virion production [20,21,22]. Since MCMV *ie-1* proteins are required for viruses to start replication from latency, the *ie-1* gene is one of the key targets for latency in animal studies. In this study, our goal is to develop an effective strategy to inhibit the growth of MCMV in cell culture and animals specifically, particularly for the removal of latent viruses.

## 2. Materials and Methods

### 2.1. Ethics Statement

For all experiments on live mice, we confirm that all methods were carried out in accordance with relevant guidelines and regulations. The protocol for all animal experiments was approved by the Institutional Animal Care and Use Committee (IACUC) of the University of California at Berkeley, USA (Protocol #R240 and #R276). All efforts were made to minimize suffering.

### 2.2. Viruses and Cell Culture

The MCMV Smith strain and mouse embryonic fibroblast NIH3T3 cells were obtained from the American Type Culture Collection (ATCC, Manassas, VA, USA). The MCMV was grown in NIH3T3 cells (ATCC) NIH3T3 cells were cultured in 500 mL Dulbecco’s modified Eagle’s medium (DMEM) (ThermFischer Scientific, Waltham, MA, USA) supplemented with 10% Nu-Serum (Coring, Union City, CA, USA), 1% Pen-Strep (100 U/mL of penicillin and 100 µg/mL of streptomycin), 1× MEM essential amino acids (EAA), 1× MEM nonessential amino acids (NEAA) and 12 mL sodium bicarbonate (ThermoFischer Scientific, USA).

### 2.3. Mice

The three-week-old immunocompetent Balb/c mice were purchased from the Jackson Laboratory, USA and used at four weeks of age.

### 2.4. Primers and Probes

MCMV M80/80.5 forward primers (5′-CTTGCCTCAGGTGCCCTCTTATTACGGAAT-3′) and reverse primers (5′- ATAAATCACACGTTCACTCCGTTAGTCCGG -3′) were both synthesized by Life technologies, Camarillo, CA, USA.

MCMV *ie-1* forward primers (5′-TCAGCCATCAACTCTGCTACCAAC-3′) and reverse primers (5′-ATCTGAAACAGCCGTATATCATCTTG-3′) were synthesized by Life technologies, USA. TaqMan probes (5′-TTCTCTGTCAGCTAGCCAATGATATCTTCGAGC-3′) were synthesized by Genscript, Nanjing, Jiangsu, China. The probe was labeled at the 5′ end with the reporter dye FAM and at the 3′ end with the quencher dye TAMRA.

### 2.5. Transcription Activator-Like Effector Nuclease (TALEN) Plasmids

Three specific pairs (left and right) of TALEN plasmids for each targeting site MCMV1-2, 3-4, 5-6, and two nonspecific pairs W1FS-W7R1, KSHV1-2 were constructed by others. They all contain a CMV promoter, a *Fok* I gene (cleavage domain) and a TALE DNA sequence using pTAL4 Leu (8467 bp, Addgene, Watertown, MA, USA) as the backbone vector. All ten TALEN plasmids are listed in Table 1.

### 2.6. Determination of the Murine Cytomegalovirus (MCMV) Growth Curve in Host Cells

NIH3T3 cells (1.00 × 10^5^ cells/well) were plated in a 12-well format containing 1 mL of growth medium and infected with MCMV (multiplicity of infection, MOI = 0.05) 1 day later (initial titer: 5.00 × 10^3^ pfu/mL). In the preliminary test, the viral titers were 4.20 × 10^3^, 1.40 × 10^5^, 5.10 × 10^5^, 2.70 × 10^5^ pfu/mL at 1, 3, 5 and 7 days post infection, respectively. We found that the viral titers were increasing for 1 to 5 days post infection, but gradually decreasing after 5 days. The results demonstrated that MCMV reached the highest titer at the 5th day post infection.

### 2.7. Determination of Transfection Efficiency

Lipofectamine™ 2000 transfection reagent was commercially obtained from Life Technologies, USA. The other transfection reagent NKS11, a new lipoid, was synthesized by others. In a safety test, the NKS11 formulation was proved to be nontoxic to Balb/c mice by weight measurement and health observation.

One day before transfection, adherent NIH3T3 cells (1.00 × 10^5^ cells/well) in 1 mL growth medium without antibiotics were plated in a 12-well format so that cells would be 90–95% confluent at the time of transfection. GFP (green fluorescent protein) plasmids (1.6 µg/well, pRK-9-Flag-EGFP, 5520 bp, Addgene, USA) were transfected into the cells to determine the transfection efficiency. The medium was changed after 4–6 h. The growth medium was removed and the cells were washed with phosphate buffered saline (PBS), followed by trypsinizing the cell pellet for 5 min using trypsin (100 µL/well) at 1, 2, and 3 days post transfection, respectively. The cell pellet was resuspended with 1 ml growth medium and cell number was counted using a hemocytometer under the fluorescence microscope.

By the data we obtained, the transfection efficiency increased during the period 1, 2, and 3 days post transfection, and then saturated at the 3rd day. The percentages were about 14.3%, 21.4%, and 21.4% using lipofectamine. The results revealed that the highest transfection efficiency for plasmids in NIH 3T3 cells was about 20–25% and it took about 2–3 days for the plasmids to transfect into cells completely. NKS 11 showed almost the same efficiency for transfection in NIH 3T3 cells.

### 2.8. Cell Count

One day before transfection, NIH3T3 cells (1.00 × 10^5^ cells/well) were plated in a 12-well format containing 1 mL of growth medium without antibiotics so that adherent cells would be 90–95% confluent at the time of transfection. TALEN plasmids MCMV 1–2, 3–4 or 5–6 were transfected for each well (1.6 µg/well, 0.8 µg for each plasmid), or none were transfected as a negative control using lipofectamine. GFP plasmids (1.6 µg/well) were transfected into the cells as a positive control. The medium was changed after 4–6 h. The growth medium was removed and the cells were washed with PBS. The cell pellet was harvested using trypsin (100 µL/well) at 1, 3, 5 and 7 days post transfection, respectively. The cell pellet was resuspended with 1 mL growth medium. We counted the cell number using a hemocytometer under a light microscope.

### 2.9. Cell Viability Assay

The viable cells were assayed using the MTT Cell Growth Assay Kit (Sigma-Alderich, Temecula, CA, USA). The trypsinized cells were appropriately diluted to adjust the cell number range of 1000–50,000 cells/well with growth medium. An amount of 0.2 mL of cell dilution was plated into each well in a 96-well format. The cells were incubated at 37 ℃ overnight. The medium was changed to make the final volume of each well 0.1 mL. We added 0.01 mL AB Solution (MTT) to each well. We incubated the cells at 37 °C for 4 h for cleavage of MTT (3-(4,5-dimethyl-2-thiazolyl)-2,5-diphenyl-2-H-tetrazolium bromide, Thiazolyl Blue Tetrazolium Bromide). We added 0.1 mL isopropanol with 0.04 N HCl to each well. The absorbance was measured on an ELISA plate reader (Bio-rad, Hercules, CA, USA) with a test wavelength of 570 nm and a reference wavelength of 630 nm within 1 h.

### 2.10. TALEN Plasmid Transfection and MCMV Infection

On the 1st day, NIH3T3 cells (1.00 × 10^5^ cells/well) in 1 mL growth medium without antibiotics were plated in a 12-well format so that adherent cells would be 90–95% confluent at the time of transfection. On the 2nd day, one pair of TALEN plasmids for each well (1.6 µg/well, 0.8 µg for each plasmid) was transfected, or none were transfected as a negative control, into the cells. GFP plasmids (1.6 µg/well) were transfected into the cells as a positive control. The medium was changed after 4–6 h. On the 3rd day, the cells were infected with MCMV (MOI = 0.05). The steps taken on the 2nd and 3rd day would be reversed if MCMV infection was prior to TALEN plasmid transfection. The medium was changed after 1 h. The cells were incubated at 37 °C in a CO_2_ incubator for 5–7 days and we changed the medium every three days.

### 2.11. Virus Titration Assay

NIH3T3 cells grown to 60–70% confluent in 12-well format were prepared for virus titration. At 1, 3, 5, and 7 days post infection, the infected cells together with the medium were harvested and followed by 10 folds of serial dilution. After 1 h of incubation with the dilution at 37 °C in a CO_2_ incubator, the prepared cells were overlaid with 2 mL fresh complete medium containing 1% low melting agarose and cultured for 4 to 5 days before the plaques were counted under a light microscope. The viral titer (pfu/mL) was determined by plaque assays. The values of the viral titers were the average of triplicate experiments.

### 2.12. Harvest of the Total DNA and Amplification of the Target DNA Sequence

The total DNA was harvested from the cell culture including the cell pellet and supernatant using the Blood and Tissue DNeasy kit (Qiagen, Germantown, MD, USA) and quantified by UV260 with a spectrophotometer. We amplified the specific product using MCMVM80/80.5 primers and the total DNA as a template by a polymerase chain reaction (PCR), under the conditions of 300 nM for each primer and 1× Hotstar Taq master mix (Qiagen, USA) in a 50 µL mixture. The thermal cycling conditions were 95 °C for 15 min followed by 35 cycles of 94 °C for 30 s, 60 °C for 30 s, 72 °C for 1 min, and 72 °C for 10 min.

### 2.13. Surveyor Nuclease Mutation Detection Assay

A Surveyor nuclease mutation detection kit (Surveyor nuclease and G + C control included) was obtained from IDT Integrated Technologies, USA [23].

We amplified wild-type (reference) and mutant (test) total DNA by PCR using MCMV M80/80.5 primers. We mixed equal amounts of reference and test PCR products. We incubated the mixture at 95 °C for 5 min in a beaker filled with 800 mL of water. We then allowed the mixture to denature in order to rehybridize, by heating and cooling it to form heteroduplexes and homoduplexes (finally leaving the water at <30 °C). We treated the annealed mixture with the Surveyor nuclease and incubated at 42 °C for 1 h. The reference PCR product was treated alone as a negative control. DNA fragments were separated by 2% agarose gel electrophoresis [23].

The cleavage efficiency of PCR products was calculated by scanning the signal strength of DNA bands on the UV illuminator. They were indicated by the percentage (%) of extra bands divided by the total bands (major bands + extra bands) in signal strength.

### 2.14. Latency Establishment, Treatment and Reactivation

In total, 27 Balb/c mice were infected with Smith strain MCMV (1 × 10^5^ pfu/mouse) intraperitoneally (IP), but 3 Balb/c mice were not infected with MCMV. In total, 30 mice were housed to establish their latency for 4–5 months [3,22,24]. We sacrificed 3 mice infected with MCMV to harvest their organs (livers, lungs, spleens, kidneys and salivary glands), in order to test whether MCMV latency was established. The remaining 24 infected Balb/c mice were divided into five experimental groups (3–5 mice/group). They were untreated or treated with TALEN plasmids by tail vein injection 8 times (once/ 5–6 days). The treatment formulation formerly confirmed to be safe for mice was as follows: for total 200 µL injection, 6 µg TALEN plasmids (0.5 µg/µL), 30 µg NKS11 (10 µg/µL), 3.125 mM Sodium Acetate (25 mM, pH5.5) and PBS in each mouse. After treatment, all 24 mice were injected with an immunosuppressive agent cyclophosphamide (Sigma-Alderich, USA) at 150 mg/kg body weight twice (1 dose/5–6 days) to reactivate latency by tail vein injection [25,26,27]. Five days later, all mice were sacrificed and their organs harvested. We sonicated the organs to harvest the homogenate and total DNA.

### 2.15. Plaque Assay of MCMV in Mouse Organs

The organ homogenates were prepared by sonication and stored in 10% skim milk at −80 °C. The concentrations of all homogenates were adjusted to 10% (100 mg/mL). The presence of infectious viruses in the livers, lungs, spleens, kidneys, and salivary glands were determined by titrating organ homogenates. Plaque assays were performed by virus titration assay. The values given were calculated as PFUs per mg of tissue.

### 2.16. Quantitative Real-Time Polymerase Chain Reaction (qPCR) Analysis for DNA Copy Number

The total DNAs were harvested in 200 µL organ homogenates (100 mg/mL) in mice tissue using a Blood and Tissue DNeasy kit. Total DNA was dissolved in 50 µL Buffer AE (Qiagen, USA).

For the generation of a standard curve, the total DNA products were amplified by PCR using MCMV *ie-1* forward and reverse primers and the total MCMV DNA as a template, under the conditions of 500 nM for each primer and 1X Hotstar Taq polymerase master mix (Qiagen, USA) in a 50 µL mixture. The thermal cycling conditions were 95 °C for 15 min followed by 43 cycles of 94 °C for 30 s, 60 °C for 30 s, 72 °C for 1 min, and 72 °C for 10 min. The MCMV *ie-1* DNA PCR product (100 bp, nucleotide no.:181091-181190 in MCMV genome) was isolated to create the DNA dilution standard using the QIAquick Gel Kit Protocol. For absolute quantification of the MCMV DNA copy number in the organs, a standard curve was generated by serial dilutions of MCMV *ie-1* DNA PCR products, such that 1 µL of the standard curve template contained 5 × 10^1^, 5 × 10^2^, 5 × 10^3^, 5 × 10^4^, 5 × 10^5^, 5 × 10^6^ for *ie-1* DNA copies. The standard curve was obtained by plotting the average threshold cycle (Ct) values against the logarithm of the target template molecules eluted from the MCMV *ie-1* DNA PCR products, followed by a sum of least squares regression analysis. Results were expressed as DNA copies/ mg of tissue. Since DNA yield per mg of tissue differs from tissue to tissue, DNA was extracted from 20 mg of each tissue and all extractions were done in triplicate and the average was used to determine the DNA yield/mg of each tissue type.

All qPCRs were performed with the TaqMan gene expression master mix (2× Hotstar Taq polymerase, Qiagen, USA) using the standard curve assay. Each sample was analyzed in triplicate at a 20 µL volume. For the *ie-1* DNA copy number assay, reaction mixtures contained 150 nM of each MCMV *ie-1* primer and 100 nM of the TaqMan probe. The amplification conditions were 50 °C for 2 min, 95 °C for 10 min, followed by 43 cycles of 95 °C for 15 s and 60 °C for 1 min. Values were calculated as copies per mg of tissue [28].

### 2.17. Statistical Analysis

The data were analyzed statistically using Microsoft Excel. In all cases, the values were the average of triplicate experiments and indicated as Mean ± SD (standard deviation).

Significant differences between the 2 groups in MCMV DNA copies in each organ of mice were determined using a two-tail Student’s t test (type 3). Data were calculated in triplicate and expressed as Mean ± SEM (standard error of the mean). To determine real-time PCR techniques and their relative sensitivity, all real-time PCR data were calculated to DNA copies/mg of tissue. In all cases, a *p* value of ≤0.05 was considered statistically significant.

## 3. Results

### 3.1. Construction of TALEN Plasmids

Three specific pairs of TALEN plasmids MCMV1-2, 3-4 and 5-6 were constructed to target the MCMV M80/80.5 overlapping region. However, the TALE DNA sequences of two nonspecific pairs of TALEN plasmids W1FS-W7R1 and KSHV1-2 were both proved to be nonhomologous to MCMV M80/80.5 targeting sites and their MCMV genome match size was less than 10 nucleotides.

Several RVDs were constructed as TALE sequences in TALEN plasmids. Each RVD recognizes one nucleotide. The RVDs of MCMV1-2, 3-4, and 5-6 were designed according to MCMV M80/80.5 ORF. MCMV M80/80.5 forward and reverse primers were designed to cover all the three pairs of targeting sites and their PCR product size was 1048 bp (Figure 2).

### 3.2. The Effect of TALEN Plasmids on Host Cells

We cotransfected TALEN plasmids MCMV1-2, 3-4 or 5-6 for each well, or did not transfect any plasmids as an untransfected (negative) control, using lipofectamine. The cell number was increasing for the first 5 days, although a slight decrease was observed on the 7th day. The growth period for NIH3T3 cells is from 1 to 5 days. All the four growth curves show almost the same trend for 7 days. TALEN plasmids MCMV1-2, 3-4 and 5-6 had no obvious effect on the growth of NIH3T3 cells compared with the untransfected control (Figure 3). Additionally, similar data were available in the cell viability assay (Table 2). These results demonstrated that the growth of cells was not influenced by the TALEN plasmids MCMV1-2, 3-4 and 5-6 for the first 5 days. TALEN plasmids exhibited no significant inhibition or enhancement on host cells.

### 3.3. The Effects of TALEN Plasmids on MCMV Titer

Host NIH3T3 cells were divided into two groups as follows: (1) TALEN plasmid transfection was prior to the MCMV infection by 1 day; (2) MCMV infection was prior to TALEN plasmid transfection by 1 day.

When the specific TALEN plasmid (MCMV1-2, 3-4 and 5-6) transfection was prior to the viral infection, we found that the viral titers decreased by about 65%, 75%, and 25%, 1, 3 and 5 days post infection, respectively, compared with the controls, using lipofectamine as a transfection reagent (Figure 4A–C). Additionally, the viral titers decreased by about 50%, 60%, and 25%, 1, 3 and 5 days post infection, respectively, compared with the controls, using NKS11 as a transfection reagent (Figure 4D–F).

However, if the viral infection was prior to the plasmid transfection, we found that TALEN plasmids resulted in no obvious growth inhibition on MCMV at 1, 3, 5 and 7 days post transfection, compared with the controls using lipofectamine or NKS11 as a transfection reagent (Figure 5).

### 3.4. Analysis of the Targeting and Cleavage of TALEN Plasmids

We explored the targeting specificity and cleavage efficiency of TALEN plasmids for MCMV using the Surveyor nuclease mutation detection assay (Figure 6) [23]. NIH3T3 cells were either treated with both lipofectamine and specific TALEN plasmids (MCMV1-2, 3-4 and 5-6), nonspecific controls (W1FS-W7R1 and KSHV1-2) or only lipofectamine as a negative control. At 5 days post infection (or transfection), we harvested their total DNA from cell culture and used the total DNA as templates to amplify their M80/80.5 PCR products. In theory, specific TALEN plasmids MCMV1-2, 3-4 and 5-6 can specifically target MCMV 80/80.5 coding sequences and cleave their PCR product (1048 bp) to produce two extra DNA bands 322 and 726 bp, 608 and 440 bp, 923 and 125 bp, respectively.

When the plasmid transfection was prior to the viral infection, the PCR products synthesized from the total DNAs targeted by TALEN plasmids were almost all the same (Figure 7A). However, we could just about clearly observe that MCMV3-4 produced two extra DNA bands (608 and 440 bp), in addition to the homoduplex bands (1048 bp). The other two specific pairs (MCMV1-2 and 5-6) did not produce dominant extra bands (only faint bands or smears were seen), which was likely due to weak bands and non-specificity of cleavage (off target). The nonspecific controls (W1FS-W7R1 and KSHV1-2) and the untransfected (negative) control did not produce any extra bands (Figure 7B). This meant that TALEN plasmids MCMV3-4 should be the most specific. The specificity is critical to avoid damaging normal and/or unrelated cells in animals; therefore, we selected MCMV3-4 as the specific treatment in the following animal studies.

However, if the viral infection was prior to the plasmid transfection, the PCR products synthesized from total DNAs targeted by TALEN plasmids were almost the same (Figure 7C). We could hardly see any extra bands on agarose gel except the positive control G + C (Figure 7D) [23]. This suggested no obvious targeting and cleavage for all PCR products. The results showed that all TALEN plasmids (specific and nonspecific) did not work on the MCMV M80/80.5 coding sequence when the viral infection was prior to the plasmid transfection.

### 3.5. Establishment of MCMV Latency in Balb/c Mice

Three Balb/c mice were not infected with MCMV and were housed for latency establishment as negative controls. They were neither treated nor reactivated during the experimental process. We could not find any plaques from all five organs homogenates and *ie*-1DNA copy numbers were not detected. The results demonstrated that there were no viruses in the mice’s organs originally and no genome in Balb/c mice is homologous to the MCMV *ie-1* gene.

For the plaque assay, it suggested no lytic viruses or only latent viruses were available in mice if the result was negative (no plaques detected). Otherwise, there were lytic viruses in mice if the result of assay was positive. For the following DNA copy number assay, if there were no DNA copies detectable this meant that there were no viruses; if the result was positive, this meant there were latent viruses in the organs.

We could not find any plaques from all five organ homogenates in 3 infected Balb/c mice, but their MCMV *ie-1* DNA copy numbers were all about 10^2^ (Table 3). The results demonstrated that there were latent viruses in the organs and MCMV latency had been established in Balb/c mice 4–5 months after MCMV infection.

### 3.6. TALEN Treatment for Balb/c Mice

In comparison with latent MCMV-infected Balb/c mice without treatment and reactivation (Table 3), the MCMV *ie-1* DNA copies of the untreated but reactivated group (Group 1 in Table 4) increased by about 5–10 folds in the livers and lungs, 3–5 folds in the spleens, 20–25 folds in the kidneys, and about 3–-5 folds in the salivary glands, respectively. This suggested that reactivation induced by the immunosuppressive agent (cyclophosphamide) takes effect to increase the viral load in mice.

In Table 4, we could not observe any plaques in the five organs in all five groups of mice. No detectable DNA copies were found in the five organs of all mice in the specific treatment group (Group 2). Despite this, the DNA copies of the untreated group (Group 1) were about 10^3^ in the livers, lungs, spleens, kidneys, and 10^2^ in the salivary glands. In the other treatment groups (Group 3, 4, 5), the DNA copies ranged from 10 to 10^2^ in the livers, lungs, spleens, kidneys, and salivary glands. However, we found that the MCMV *ie-1* DNA copies of the untreated group (Group 1) were more than those of the less specific treatment groups (Group 3, 4) and the nonspecific treatment group (Group 5) by about 10–100 folds in the livers, lungs, spleens, kidneys, and by about 3–10 folds in the salivary glands. Overall, we could conclude that the specific treatment group (Group 2) was the most efficient one to remove viral load in mouse organs.

## 4. Discussion

The transfection efficiency of plasmids in NIH3T3 cells was about 20–25% in cell culture. We realized that an elevated level of transfection efficiency could increase the efficacy in inhibiting virus growth in cell culture. Despite this, we did not sort the transfected cells using flow cytometry for the studies in cell culture, because it was not feasible for us to do so in mice. To establish a more similar animal model, we used all the cells including untransfected and transfected cells for our ex vivo studies. To enhance the efficacy, multiple round injections for the formulation transfection during the treatment period are required in animal studies.

Cultured NIH3T3 cells are not specific for latency studies. If the viral infection is prior to the plasmid transfection, NIH3T3 cells are vulnerable to viruses because they do not have the same immune system as animals. In this case, TALENs can hardly protect the host cells from the viral infection, because the viral titer increases rapidly to about 10^5^ pfu/mL within 1–3 days. However, if the plasmid transfection is prior to the viral infection, TALEN plasmid copy number might have increased and induced innate immune responses of host cells to secret cytokine or other factors to fight against invading viruses. Therefore, TALENs could inhibit virus growth by about 50–75% when the viral titer was 10^3^–10^4^ pfu/mL, and about 25% when the viral titer was about 10^5^ pfu/mL.

In cell culture, the results for specific target and cleavage efficiency of TALEN plasmids reveal that some of the plasmids work well on MCMV M80/80.5 target sites. Specific TALEN plasmids also demonstrated their effects on the inhibition of virus growth, ranging from 25–75% depending on the viral titer. Although the decreasing level of viral titers was varied under different conditions, they had the same trend for the inhibition of virus growth. We also found that the higher the viral titer, the lower the effect of the TALEN plasmids on virus growth. Our findings indicated that the inhibition effect on MCMV is about 20–25% when the viral titer reaches the highest level (10^5^). The reasons might be that the amino acid-nucleotide recognition frequency is not absolute (e.g., NI-A: 55%, NG-T: 50%, NN-G: 7%, HD-C: 69%) and the transfection efficiency is about 20–25% in cell culture.

The specificity, efficiency and biosafety of delivery tools are critical for drug delivery in animal studies and human clinical trials [29]. Lipofectamine, one of the most common transfection reagents used in cell culture, is known to be toxic to animals. Our results have shown that the new transfection reagent NKS11 can work almost as well as lipofectamine for inhibiting virus lytic replication in cell culture, when TALEN plasmid transfection is prior to MCMV infection. Fortunately, NKS11 also proved to be nontoxic to Balb/c mice in the preliminary tests.

The viral *ie-1* DNA copies increased once Balb/c mice were infected with MCMV. An absence of plaques meant that there were no viruses during the lytic cycle. In latency, plaques were not detectable and the *ie-1* DNA copy number was low but detectable. The *ie-1* DNA copies increased once latency was reactivated. In Table 4, the viral load significantly increased after reactivation in all five organs of mice if there was no treatment (Group 1), compared with the treatment groups (Group 2–5). After treatment, the latent MCMV was removed by TALENs, so the viral load was undetected or significantly decreased even though latency was reactivated using an immunosuppressive agent (Group 2–5). Additionally, if the TALENs are only targeting the reactivating viruses and not the latent pools, it is impossible for the specific treatment group (Group 2) to be all ND (not detected) in the *ie-1* DNA copy number assay for all five organs of mice. Thus, we consider that it is possible to remove latent viruses using this strategy.

In animal studies, the less specific and nonspecific treatment groups (Group 3, 4, 5) also worked well in reducing viral load, although they were less efficient than the specific treatment group (Group 2). This might be explained as follows. Firstly, during the multiple-round injection for TALEN plasmid treatment, recognition of foreign DNA in intracellular compartments or in the cytoplasm of host cells sends a signal of pathogenic invasion. In response, the innate immune DNA-sensing pathways start an antimicrobial type I interferon (IFN) (mainly IFNα and β) secretion to act against viruses. Acting in paracrine or autocrine models, IFNs stimulate intracellular and intercellular networks for regulating innate and acquired immunity in mice that are resistant to the viral infections [30,31,32,33,34,35]. However, there are no data that categorically show that the latent virus has been eradicated and that this is an IFN effect in our studies. Secondly, there are three possible mechanisms for TALEN plasmids to work on DNA, namely cleavage of target DNA, induction of DNA mutation, and inhibition of DNA transcription and translation [17]. Thirdly, treatment is very complex in animals; it is also influenced by other factors such as the individual diversity of mice and the efficiency of tail vein injection.

Currently, it is impossible to eradicate latent CMV viruses in animals, although there are some effective drugs (e.g., Ganciclovir, Valganciclovir, and Foscarnet) for the treatment of active infections. Our data indicated that TALEN plasmids which could specifically target and cleave MCMV M80/80.5 ORF would effectively reduce the viral load in Balb/c mice, so that they resulted in the implicative decrease of *ie-1* DNA copies level. The viral latent infection in humans is a major barrier for effective treatment and also a long-term risk to the host. Although the mechanisms for inhibiting MCMV are still poorly understood, our studies demonstrate that the removal of latent MCMV in animals is possible using TALENs.

## Figures and Tables

**Figure 1 viruses-11-00414-f001:**
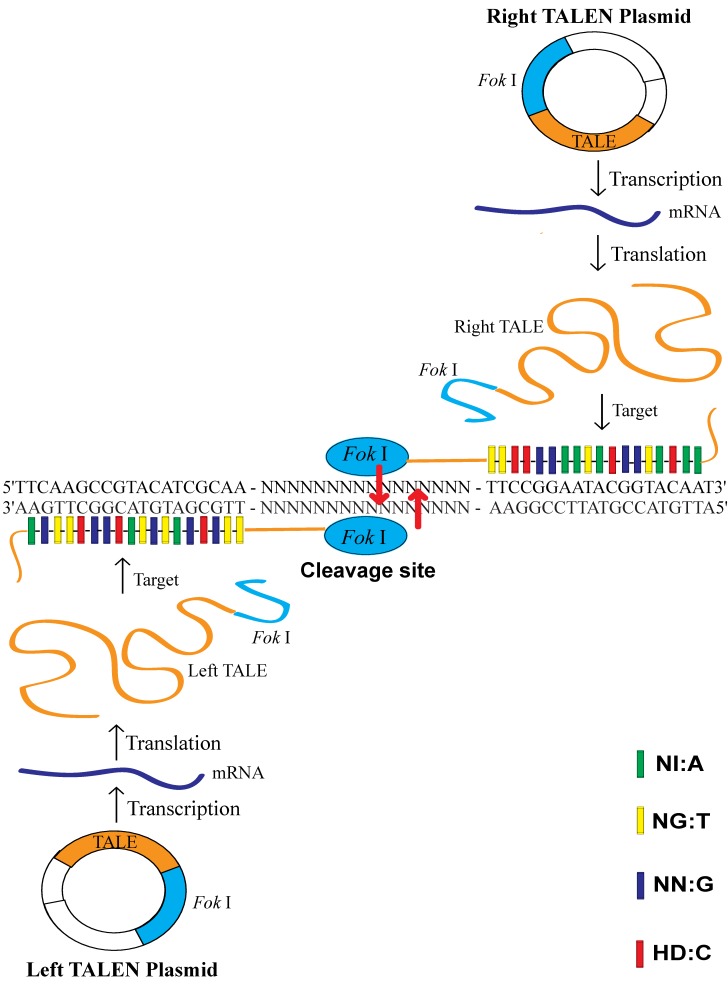
Transcription activator-like effector nuclease (TALEN) targeting and cleavage on the target site. The specific repeat variable di-residue (RVD) used to recognize each base is indicated by shading, as defined in the key (NI: A, NG: T, NN: G, HD: C). The cleavage site is in the adjoining region between the left and right target site. The left transcription activator-like effector (TALE) and the right TALE of TALEN plasmids recognized their target sequence and allowed their associated *Fok* I endonucleases to work as homodimers to cleave the sense strand 9 bp and antisense strand 13 bp downstream of the binding site. The binding of the TALEs to the target sites allows *Fok* I to dimerize and create a double-stranded break (DSB) with sticky ends within the spacer.

**Figure 2 viruses-11-00414-f002:**
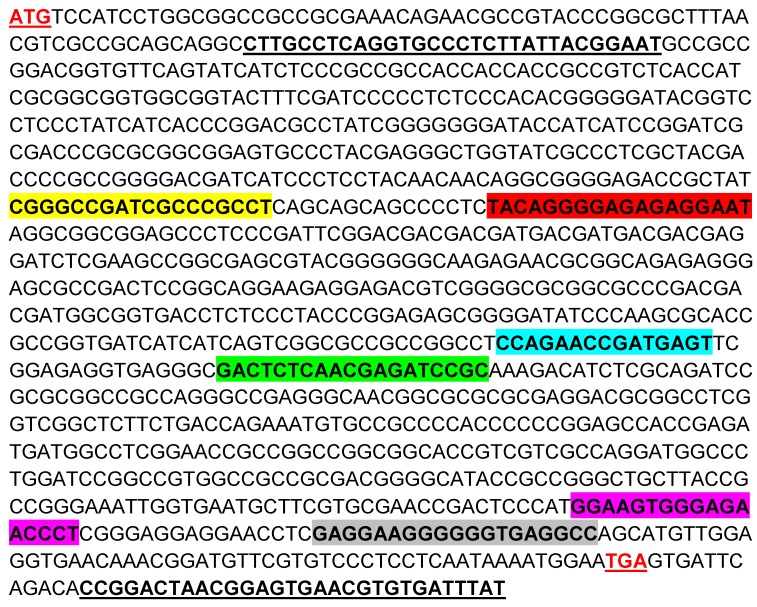
Targeting sites in the MCMV M80 and M80.5 overlapping region (M80/80.5, nucleotide no. 114434-115507). The targeting sites of TALEN plasmids MCMV 1, 2, 3, 4, 5 and 6 are highlighted in yellow, red, blue, green, purple and gray, respectively. ATG (start codon for M80.5) and TGA (stop codon for M80 and M80.5) are both in red and underlined. MCMV M80/80.5 forward and reverse primers are bolded and underlined.

**Figure 3 viruses-11-00414-f003:**
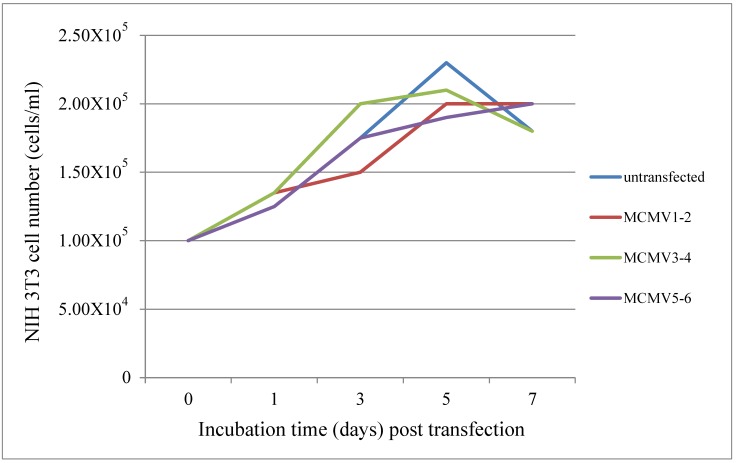
Host cell growth under the effect of TALEN plasmids. NIH 3T3 cells (1.00 × 10^5^ cells/mL) were transfected with TALEN plasmids MCMV1-2, 3-4 or 5-6, respectively. The total cells were harvested and counted 1, 3, 5 and 7 days post transfection, respectively.

**Figure 4 viruses-11-00414-f004:**
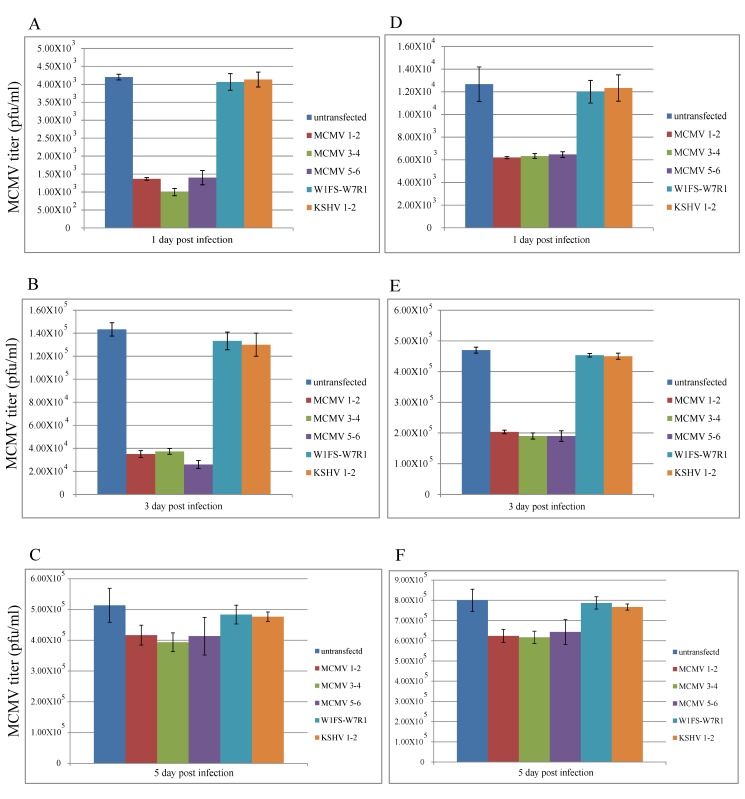
TALEN plasmid transfection was prior to MCMV infection. When using lipofectamine as a transfection reagent, the viral titers were determined 1, 3 and 5 days post infection, respectively, in (**A**–**C**). When using NKS11 as a transfection reagent, the viral titers were determined 1, 3 and 5 days post infection, respectively, in (**D**–**F**). All the data are expressed by columns (mean ± standard deviation).

**Figure 5 viruses-11-00414-f005:**
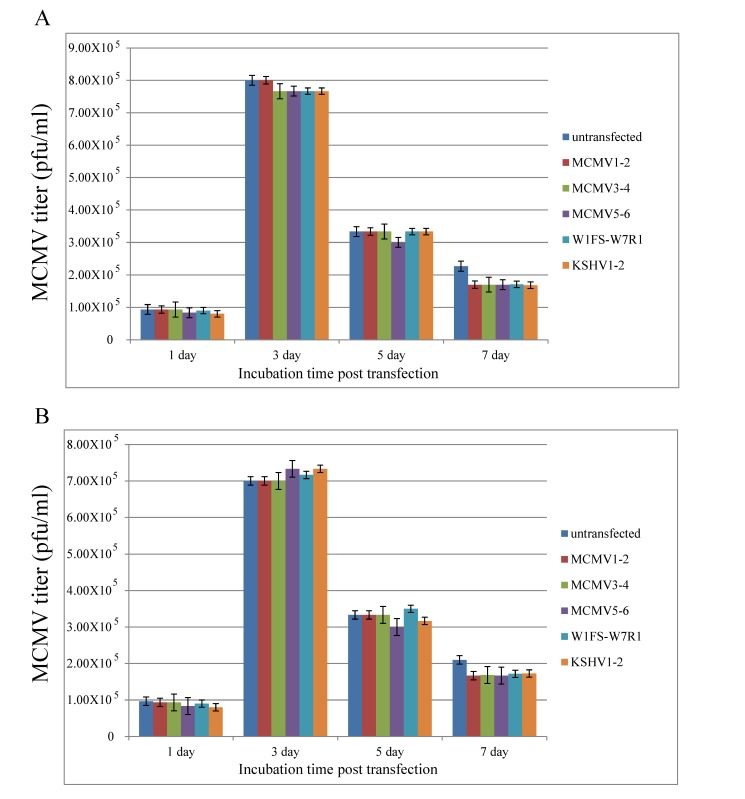
MCMV infection was prior to TALEN plasmid transfection. Viral titers were determined at 1, 3, 5 and 7 days post transfection using lipofectamine (**A**) or NKS11 (**B**), respectively. All the data are expressed by columns (mean ± standard deviation).

**Figure 6 viruses-11-00414-f006:**
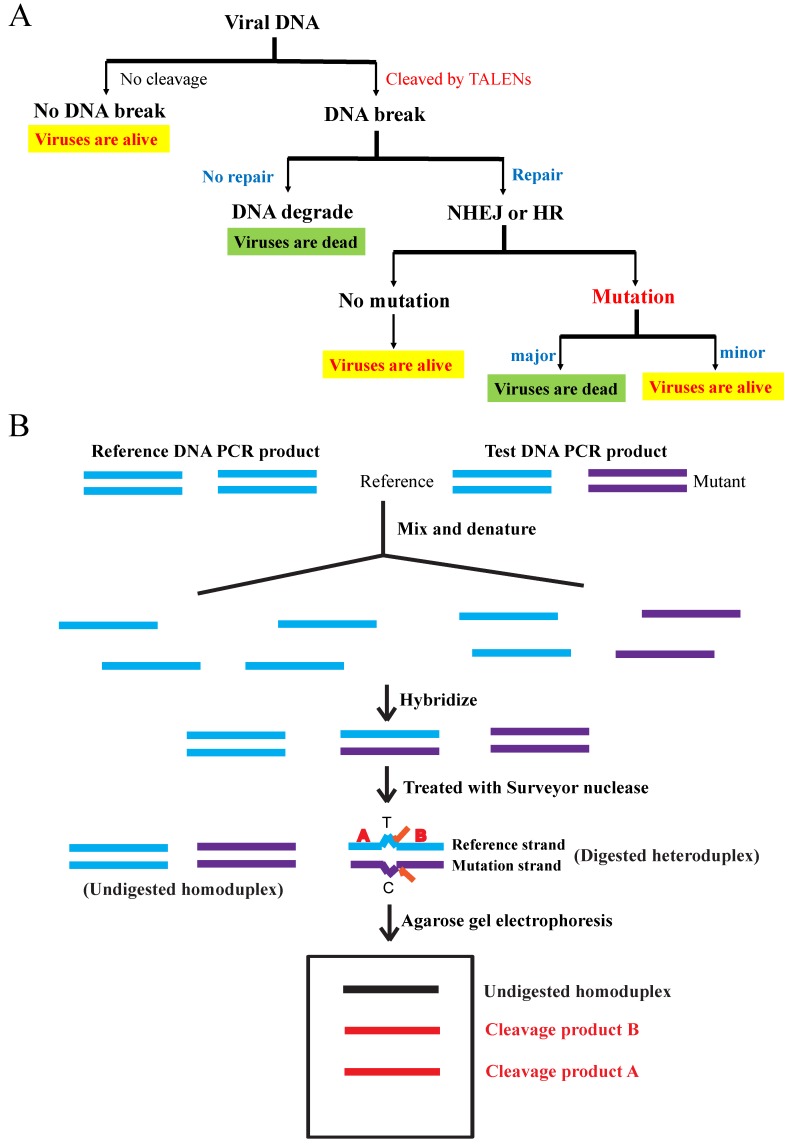
(**A**) Viral DNA targeting and cleavage: Once TALENs target and cleave the viral DNA, they will create a double stranded break (DSB). Either repaired by homologous recombination (HR) or non-homologous end joining (NHEJ), a mild mutation such as mismatch may appear. If viral DNA is mutated seriously or not repaired, viruses will be dead (highlighted in green). Viruses without cleavage, viruses without mutations, and viruses only with mild mutations are all alive (highlighted in yellow). (**B**) Surveyor nuclease mutation detection assay: Total DNA of all alive viruses are harvested and amplified by the polymerase chain reaction (PCR), followed by the denaturation, hybridization and analysis of the PCR products. Surveyor nuclease is a mismatch-specific endonuclease.

**Figure 7 viruses-11-00414-f007:**
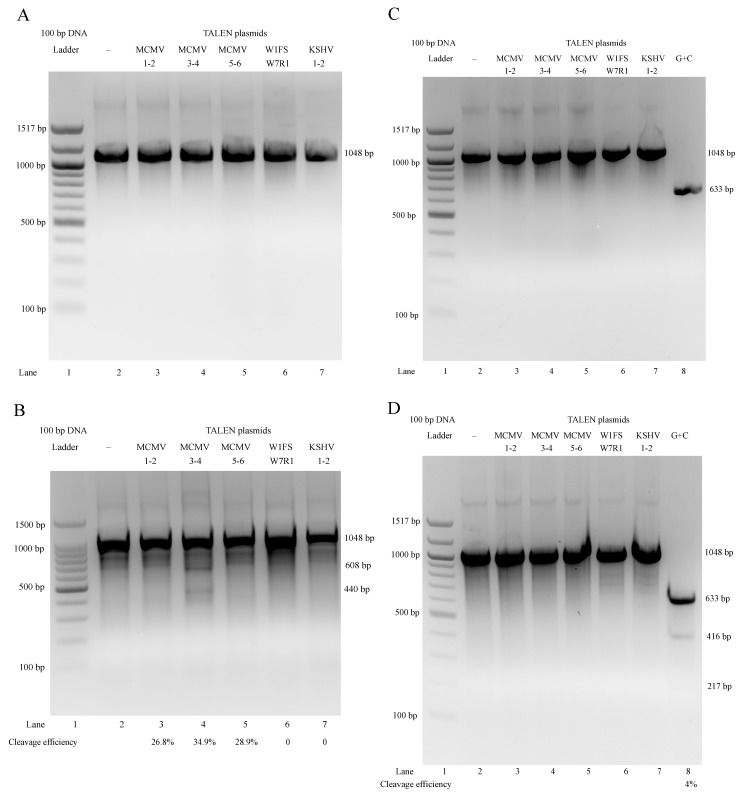
TALEN plasmid transfection was prior to MCMV infection: (**A**) Only homoduplex (or heteroduplex) PCR products (1048 bp) were observed (Lane 2–7). (**B**) Only MCMV3-4 (Lane 4) target and cleave PCR products produced two extra DNA bands (608 and 440 bp) clearly. MCMV1-2 and 5-6 (Lane 3 and 5) could not produce dominant extra bands, compared with the negative control (Lane 2). Neither the nonspecific TALEN plasmids W1FS-W7R1 (Lane 6) nor KSHV1-2 (Lane 7) could produce any extra bands, the same as the negative control (Lane 2). MCMV infection was prior to TALEN plasmid transfection: (**C**) Only homoduplex (or heteroduplex) PCR products (1048 bp) were observed (Lane 2-7). (**D**) No extra bands produced by TALEN plasmids were observed (Lane 3–7). Only positive control G + C (Lane 8) could produce dominant extra bands (416 and 217 bp) in addition to a major band (633 bp), compared with the negative control (Lane 2).

**Table 1 viruses-11-00414-t001:** List of TALEN plasmids.

TALEN Plasmid	Target DNA Sequence	Nucleotide No.	Target, Gene
MCMV 1	CGGGCCGATCGCCCGCCT	18	MCMV, M80/80.5
MCMV 2	TACAGGGGAGAGAGGAAT	18	MCMV, M80/80.5
MCMV 3	CCAGAACCGATGAGT	15	MCMV, M80/80.5
MCMV 4	GACTCTCAACGAGATCCGC	19	MCMV, M80/80.5
MCMV 5	GGAAGTGGGAGAACCCT	17	MCMV, M80/80.5
MCMV 6	GAGGAAGGGGGGTGAGGCC	19	MCMV, M80/80.5
W1FS	GCTGATTCTTCCCTGTG	17	293T cell, WAS
W7R1	AAGAGTGGATGGAGG	15	293T cell, WAS
KSHV 1	TTACAATGGTGTAGGTG	17	KSHV, RTA
KSHV 2	AGCTCTACGTCCGAAC	16	KSHV, RTA

Abbreviations: MCMV (murine cytomegalovirus), M80/80.5 (the overlapping region of MCMV M80 and M80.5), 293T cell (human embryonic kidney 293 cells), WAS (a specific gene in 293 cells), KSHV (Kaposi’s sarcoma-associated herpes virus), RTA (replication and transcription activator gene in KSHV).

**Table 2 viruses-11-00414-t002:** Cell viability assay (initial NIH 3T3 cell count: 1.00 × 10^5^ cells/mL).

Day Post Transfection	Untransfected	MCMV1-2	MCMV3-4	MCMV5-6
1	1.20 × 10^5^ ± 5000	1.25 × 10^5^ ± 5000	1.27 × 10^5^ ± 2890	1.17 × 10^5^ ± 5700
3	1.68 × 10^5^ ± 2890	1.42 × 10^5^ ± 2890	1.92 × 10^5^ ± 7640	1.70 × 10^5^ ± 5000
5	2.25 × 10^5^ ± 5000	1.92 × 10^5^ ± 7640	2.03 × 10^5^ ± 5770	1.88 × 10^5^ ± 2890
7	1.72 × 10^5^ ± 2890	1.87 × 10^5^ ± 5770	1.73 × 10^5^ ± 5770	1.85 × 10^5^ ± 5000

The cell number is expressed by cells/mL, and all data are indicated as Mean ± SD (standard deviation).

**Table 3 viruses-11-00414-t003:** Assay of MCMV-infected mouse organs for latency establishment.

Organ	Liver	Lung	Spleen	Kidney	Salivary Gland
Plaque assay	ND	ND	ND	ND	ND
DNA copies	315 ± 315	535 ± 94	226 ± 174	86 ± 86	67 ± 67

Balb/c mice were infected with viruses, but neither treated nor reactivated. Plaque assay (pfu/mg of tissue): all five organs from three mice were tested. Copy number of *ie-1* (DNA copies/mg of tissue): Mean ± SEM, $\mathrm{SEM}=\mathrm{SD}/\sqrt{n}$, n = 3; SEM: standard error of mean; SD: standard deviation; n: sample size; ND: not detected (less than 10 pfu/mL for plaque assay, less than 50 copies/mg tissue for DNA copy number assay).

**Table 4 viruses-11-00414-t004:** Assay of MCMV-infected mouse organs for TALEN treatment.

Group	1 (*n* = 3)	2 (*n* = 4)	3 (*n* = 4)	4 (*n* = 5)	5 (*n* = 5)
Plaque Assay	ND	ND	ND	ND	ND
Organ	DNA Copies				
Liver	2284 ± 1301	ND (*p* = 0.0004)	32 ± 32 (*p* = 0.0005)	25 ± 25 (*p* = 0.0005)	ND (*p* = 0.0004)
Lung	3777 ± 2865	ND (*p* = 0.0001)	142 ± 82 (*p* = 0.0008)	353 ± 168 (*p* = 0.0032)	311 ± 153 (*p* = 0.0041)
Spleen	1073 ± 587	ND (*p* = 0.0041)	ND (*p* = 0.0052)	ND (*p* = 0.0041)	29 ± 29 (*p* = 0.0087)
Kidney	2033 ± 1413	ND (*p* = 0.0046)	ND (*p* = 0.0046)	134 ± 82 (*p* = 0.0309)	22 ± 22 (*p* = 0.0068)
Salivary gland	379 ± 260	ND (*p* = 0.0158)	97±57 (*p* = 0.2470)	172 ± 127 (*p* = 0.2768)	38 ± 38 (*p* = 0.0468)

*p* value: comparison of the untreated group (Group 1) versus the treatment groups (Group 2, 3, 4 and 5) respectively; Plaque assay (pfu/mL): five organs from all five groups of mice were tested; Copy number of *ie-1* (DNA copies/mg of tissue): Mean ± SEM, $\mathrm{SEM}=\mathrm{SD}/\sqrt{n}$; SEM: standard error of mean; SD: standard deviation; n: sample size; ND: not detected (less than 10 pfu/mL for plaque assay, less than 50 copies/mg tissue for DNA copy number assay); Experimental groups were all reactivated by cyclophosphamide; Group 1: No treatment (only treated with phosphate buffered saline); Group 2: specific treatment (TALEN plasmids MCMV3-4) with transfection reagent NKS11; Group 3: specific treatment (TALEN plasmids MCMV3-4) without transfection reagent NKS11; Group 4: less specific treatment (TALEN plasmids MCMV1-2, 3-4 and 5-6) with transfection reagent NKS11; Group 5: nonspecific treatment (TALEN plasmids KSHV3-4) with transfection reagent NKS11.

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
