# Peer review of "Potential Application of TALENs against Murine Cytomegalovirus Latent Infections"

_viruses, 2019, doi:10.3390/v11050414_

Round 1
Reviewer 1 Report
In this manuscript, Chen and Chen report on the “clearance of latent murine cytomegalovirus [mCMV] infections using TALENs”. The authors generated three pairs of TALEN plasmids directed at the sequence encoding the M80 and M80.5 proteins in the mCMV genome. They demonstrate successful targeting of M80/80.5 following TALEN transfection and corresponding inhibition of mCMV replication in NIH3T3 cells. One pair of TALEN plasmids was further tested by injection into BALB/c mice infected with mCMV. The TALEN injections resulted in undetectable levels of mCMV DNA in several organs, suggesting that latent viral loads were diminished.
To my knowledge, this is the first report about the use of TALENs to target CMV. The study appears to be technically sound, and most conclusions seem to be justified by the data although effects tend to be moderate. That said, the manuscript is flawed by major shortcomings in data presentation that have to be addressed.
1. The way the figures are presented may be acceptable in a student thesis, but not in a scientific paper. Bar charts (Figures 5 to 8) should be presented as composite figures rather than individual graphs. The same applies to the gel images shown in Figures 10 and 11. Some of the diagrams do not add much and may be removed. Generally, the number of figures should be reduced from 13 to a much smaller number of composite figures.
2. There are language issues, including many grammatical errors, throughout the text that make reading cumbersome. The text requires substantial language editing by an English proficient person.
3. More importantly, the text comes with factual errors including, but not limited to:
- The Abstract states that “current therapies for CMV latent infections are limited in efficacy” (line 10-11). However, there are no current therapies for latent CMV infection and therapies for active infection are limited by side effects and other problems rather than efficacy.
- The authors claim that mCMV and hCMV proteins are 70% identical (line 35), which is clearly not the case.
- It is not clear to me what “most of the different coding proteins are glycoproteins” (line 35-36) is supposed to mean. I do not believe that most mCMV or hCMV proteins are glycoproteins.
- Revise sentence ending in “which genes are stably floating in the cytoplasm or nucleus as distinct objects” (line 97-98). CMV genes do not float as distinct objects, not even in the nucleus.
- Revise “NIH3T3 cells […] have no immune system” (line 323), and “virus replication is much faster than plasmid replication” (line 324). The Discussion contains similar sentences on replicating plasmids. NIH3T3 cells do have innate immune responses. The plasmids transfected in this study do not replicate, or do they?
4. The title is an overstatement, since the authors cannot be sure that latent mCMV has been cleared from the animals using TALENs even if qPCR for viral DNA is negative.
5. Some background on the mCMV M80/80.5 genes and proteins, including their importance in the viral life cycle, should be provided in the Introduction. Why of the many mCMV genes did the authors pick M80/80.5, and why exactly the region overlapping the two coding sequences?
6. The structure of Materials and Methods with its 19 different subsections resembles that of a student thesis rather than a research paper. Some of these subsections should be consolidated. For example, the subsections on “Transfectants” (line 127), which should actually read “Transfection reagents” and “Determination of transfection efficiency for plasmids” (line 165) should be combined. Likewise, it does not make sense to have separate subsections on “Mutation detection kit” (line 133) and “Surveyor nuclease mutation detection assay” (line 202). Generally, materials are not listed separately from methods in research papers (with some exceptions). The standard curve for qPCR (line 236) and the actual qPCRs (line 249) should also be in the same subsection.
7. May Table 1 be better placed in the Materials and Methods section?
8. Section 3.2 including Figure 3 may be dispensable. The relevant information provided with this figure is minimal.
9. The data in section 3.3 including Figure 4 would be strengthened by adding results from a cell viability assay.
10. Section 3.4 seems to be dispensable, especially since the primary GFP data are not shown. The numbers could be included in Materials and Methods or other Results sections instead. Are there numbers for NKS11 as well? It is also unclear why this section is placed after and not before the first transfection experiment (section 3.3).
11. How was the cleavage efficiency in Figure 10B calculated? I do not see any 608 and 440 bp bands in the MCMV 1-2 and MCMV 5-6 lanes.
12. The animal experiments are not properly introduced. It is not appropriate to start the description of these experiments with “The standard curve for ie-1 DNA copy number assay” (line 423), which reads more like Materials and Methods anyway. Subsections 3.7, 3.8 and 3.9 (and perhaps also 3.10) should be combined into one section. The standard curve (Figure 12) does not warrant a separate figure. Likewise, Table 2 should be deleted and information incorporated in the main text.
13. Revise sentence “For the plaque assay, no plaques were detectable hinted no viruses or only with latent viruses”.
14. Have the authors determined the sensitivity of their qPCR assay?
15. Figures 13 and 14 appear to be dispensable.
16. It is unclear why the authors believe that transfecting the cells first and infecting them after that “might be more similar to the latency model of viruses” (line 325) than infecting first and transfecting subsequently. What does “We need to have a test that the TALEN plasmids transfection was prior to the MCMV infection” (line 325-326) mean? Combine 3.5, 3.5.1 and 3.5.2 into one section (no separate headings for 3.5.1 and 3.5.2)? Reduce number of figures: combine all graphs with transfection before infection (Figure 5, Figure 7) in one figure and all graphs with transfection after infection (Figure 6, Figure 8) in another figure?
17. If the TALEN pair MCMV 3-4 is more efficient in targeting/cleaving M80/80.5 (Figure 10), why does this not result in more efficient inhibition of viral replication (Figure 5, Figure 7)?
18. Discuss potential explanations for different efficacy of TALENs between transfection pre- or post-infection?
19. Discuss innate immune (foreign DNA) response triggered by plasmid injection that may account for non-specific antiviral effects of TALENs.
20. Remove numbers from section/subsection headings.
Author Response
Responses to the review report
Reviewer 1:
In this manuscript, Chen and Chen report on the “clearance of latent murine cytomegalovirus [mCMV] infections using TALENs”. The authors generated three pairs of TALEN plasmids directed at the sequence encoding the M80 and M80.5 proteins in the mCMV genome. They demonstrate successful targeting of M80/80.5 following TALEN transfection and corresponding inhibition of mCMV replication in NIH3T3 cells. One pair of TALEN plasmids was further tested by injection to BALB/c mice infected with mCMV. The TALEN injections resulted in undetectable levels of mCMV DNA in several organs, suggesting that latent viral loads were diminished.
To my knowledge, this is the first report about the use of TALENs to target CMV. The study appears to be technically sound, and most conclusions seem to be justified by the data although effects tend to be moderate. That said, the manuscript is flawed by major shortcomings in data presentation that have to be addressed.
Response;
We thank the reviewers’ comment for helping us to improve this manuscript. We will try our best to address the presented data to overcome the major shortcoming. Attachment is our revised manuscript.
The way the figures are presented may be acceptable in a student thesis, but not in a scientific paper. Bar charts (Figures 5 to 8) should be presented as composite figures rather than individual graphs. The same applies to the gel images shown in Figures 10 and 11. Some of the diagrams do not add much and may be removed. Generally, the number of figures should be reduced from 13 to a much smaller number of composite figures.
Response:
We have combined original Figure 5-8 to two composite Figure 4 (P.12, Line 1-6) and 5 (P.13, Line 1-4), original Figure 10-11 to composite Figure 7 (P.16, Line 1-12), respectively.
There are language issues, including many grammatical errors, throughout the text that make reading cumbersome. The text requires substantial language editing by an English proficient person.
Response:
We have checked the vocabulary, grammar, and sentence structure once more carefully and ask for the other’s help in extensively editing of English language. We removed a lot of cumbersome words and sentences to make it more comprehensive and easier to read all over the manuscript. Additionally, we rearranged the content of this manuscript to make it more logical and organized.
3. More importantly, the text comes with factual errors including, but not limited to:
- The Abstract states that “current therapies for CMV latent infections are limited in efficacy” (line 10-11). However, there are no current therapies for latent CMV infection and therapies for active infection are limited by side effects and other problems rather than efficacy.
Response:
We have revised this sentence to “Currently, there are no therapies for latent CMV infection and therapies for active infection are limited by side effects and other problems.” (P1, Line 10-12)
- The authors claim that mCMV and hCMV proteins are 70% identical (line 35), which is clearly not the case.
Response:
We are sorry to have an overstatement and removed “Among their coding proteins, about 70% are identical (P.1).
- It is not clear to me what “most of the different coding proteins are glycoproteins” (line 35-36) is supposed to mean. I do not believe that most mCMV or hCMV proteins are glycoproteins.
Response:
We are sorry to have an overstatement and removed “most of the different coding proteins are glycoproteins on the envelope for recognition.”(P.1)
- Revise sentence ending in “which genes are stably floating in the cytoplasm or nucleus as distinct objects” (line 97-98). CMV genes do not float as distinct objects, not even in the nucleus.
Response:
We have revised the sentence to “CMV belongs to the episomal latency model which was essentially quiescent in myeloid progenitor vells, and is reactivated by differentiation, inflammation, immunosuppression or critical diseases..” (P.4, Line 23-25). Additionally, a new reference 20 is inserted (P.21, P.36-37).
- Revise “NIH3T3 cells […] have no immune system” (line 323), and “virus replication is much faster than plasmid replication” (line 324). The Discussion contains similar sentences on replicating plasmids. NIH3T3 cells do have innate immune responses. The plasmids transfected in this study do not replicate, or do they?
Response:
We have removed the sentence “virus replication is much faster than plasmid replication” and the paragraph “NIH3T3 cells […] have no immune system” in the manuscript. Additionally, we have added a paragraph “Cultured NIH3T3 cells are not specific for latency studies. If viral infection is prior to plasmid transfection, NIH3T3 cells are vulnerable to viruses because they don’t have the same immune system as animals...” in the “Discussion” section (P.19, Lin 4-11) to discuss them.
We are sorry that we do not have data to show if the plasmids transfected replicate or not in this study.
The title is an overstatement, since the authors cannot be sure that latent mCMV has been cleared from the animals using TALENs even if qPCR for viral DNA is negative.
Response:
We have revised our title to “Potential Application of TALENs against Murine Cytomegalovirus Latent Infections”. (P.1, Line 2-3)
Some background on the mCMV M80/80.5 genes and proteins, including their importance in the viral life cycle, should be provided in the Introduction. Why of the many mCMV genes did the authors pick M80/80.5, and why exactly the region overlapping the two coding sequences?
Response:
We have added a paragraph in the “Introduction” section to explain why we picked MCMV M80/80.5 as a TALENs target site (P.1, Line 40- P.2, Line 10).
The structure of Materials and Methods with its 19 different subsections resembles that of a student thesis rather than a research paper. Some of these subsections should be consolidated. For example, the subsections on “Transfectants” (line 127), which should actually read “Transfection reagents” and “Determination of transfection efficiency for plasmids” (line 165) should be combined. Likewise, it does not make sense to have separate subsections on “Mutation detection kit” (line 133) and “Surveyor nuclease mutation detection assay” (line 202). Generally, materials are not listed separately from methods in research papers (with some exceptions). The standard curve for qPCR (line 236) and the actual qPCRs (line 249) should also be in the same subsection.
Responses:
We have changed “transfectant” to “transfection reagent” all over the manuscript.
In the ‘Materials and Methods” section, related materials and method have been listed together. For example, we have combined the original subsection “Transfectant” and “Determination of transfection efficiency for plasmids” to “2.7. Determination of transfection efficiency” (P.6, Line 9-27), the original section “Mutation detection kit” and “Surveyor nuclease mutation detection assay” to “2.13. Surveyor nuclease mutation detection assay” (P.7, Line 36-P.8, Line 5), the original section “Quantification of MCMV DNA for a standard curve” and “Quantitative real-time PCR (qPCR) analysis for DNA copy number” to “2.16. Quantitative real-time PCR (qPCR) analysis for DNA copy number” (P.8, Line 29-P.9, Line 10).
In the “Results” section, we have combined the original subsection 3.7-3.9 to “3.7. Establishment of MCMV latency in Balb/c mice” (P.16, Line 13-P.17, Line 24).
May Table 1 be better placed in the Materials and Methods section?
Response:
Yes, we have moved “Table 1” to the “Materials and Methods” section (P.5, Lin 28-32).
Section 3.2 including Figure 3 may be dispensable. The relevant information provided with this figure is minimal.
Response:
We have removed the original Figure 3 and described the information we got in the subsection “2.6. Determination of MCMV growth curve in host cells” (P.6, Line 1-7).
The data in section 3.3 including Figure 4 would be strengthened by adding results from a cell viability assay.
Response:
We have added a new subsection “2.9. Cell viability assay” (P.6, Line 40-P.7, Line 6) and provided the data in the subsection “3.3. The effect of TALEN plasmids on host cells” (Table 2, P.11, Line 5-7).
Section 3.4 seems to be dispensable, especially since the primary GFP data are not shown. The numbers could be included in Materials and Methods or other Results sections instead. Are there numbers for NKS11 as well? It is also unclear why this section is placed after and not before the first transfection experiment (section 3.3).
Response:
We have removed the original section 3.4 and included the related information in the subsection “2.7. Determination of transfection efficiency” (P.6, Line 9-27).
How was the cleavage efficiency in Figure 10B calculated? I do not see any 608 and 440 bp bands in the MCMV 1-2 and MCMV 5-6 lanes.
Response:
We have added a statement in 2.13. Surveyor nuclease mutation detection assay “The cleavage efficiency of PCR products was calculated by scanning signal strength of DNA bands on the UV illuminator. They were indicated by the percentage (%) of extra bands signal strength divided by the total bands (major bands + extra bands) signal strength.” (P.8, Line 3-4).
The extra band produced by MCMV 1-2 and MCMV 5-6 should be 322 and 726 bp; 923 and 125 bp, respectively (P.13, Line 13-P.14, Line 2). The reason why they are not visible on the gel may be the extra band is very weak and/or the cleavage is not very specific (off target). We can clearly see the 608 and 440 bp extra bands produced by MCMV 3-4, because this one is more specific than MCMV 1-2 and MCMV 5-6 (P.15, Line 10-24).
The animal experiments are not properly introduced. It is not appropriate to start the description of these experiments with “The standard curve for ie-1 DNA copy number assay” (line 423), which reads more like Materials and Methods anyway. Subsections 3.7, 3.8 and 3.9 (and perhaps also 3.10) should be combined to one section. The standard curve (Figure 12) does not warrant a separate figure. Likewise, Table 2 should be deleted and information incorporated in the main text.
Response:
We have combined Sections 3.7, 3.8 and 3.9 to a new section “3.7. Establishment of MCMV latency in Balb/c mice” (P.16, Line 13-P.17, Line 24) and deleted Table 2. Additionally, we have deleted original Figure 12 and incorporated their information in the main text, respectively (P.7, Line 36-P.8, Line 5).
Revise sentence “For the plaque assay, no plaques were detectable hinted no viruses or only with latent viruses”.
Response:
We have revised this paragraph to “For the plaque assay, it hinted no lytic viruses or only latent viruses available in mice if the result were negative (no plaques detectable). Otherwise, there were lytic viruses in mice if the result of assay was positive. For the following DNA copy number assay, there were no DNA copies detectable meant no viruses really; nevertheless, this meant there were latent viruses in the organs if the result was positive.” (P.17, Line 6-10).
14. Have the authors determined the sensitivity of their qPCR assay?
Response:
Every time when we performed qPCR assay, a standard curve using serial dilutions 5X101, 5X102, 5X103, 5X104, 5X105, 5X106 /µl for ie-1 DNA copies were used for calibration. Therefore, we set the minimal sensitivity is 50 ie-1 DNA DNA copies for all cases. If the value is lower than 50 DNA copies, we indicated it as ND (not detected). The data shown in Table 4 are Mean±SEM ( ie-1 DNA copies/mg of tissue), so the values may be lower than 50 (P.8, Line 33-P.9, Line 4).
15. Figures 13 and 14 appear to be dispensable.
Response:
We have removed original Figure 13 and 14.
16. It is unclear why the authors believe that transfecting the cells first and infecting them after that “might be more similar to the latency model of viruses” (line 325) than infecting first and transfecting subsequently. What does “We need to have a test that the TALEN plasmids transfection was prior to the MCMV infection” (line 325-326) mean? Combine 3.5, 3.5.1 and 3.5.2 to one section (no separate headings for 3.5.1 and 3.5.2)? Reduce number of figures: combine all graphs with transfection before infection (Figure 5, Figure 7) in one figure and all graphs with transfection after infection (Figure 6, Figure 8) in another figure?
Response:
We are sorry to have an overstatement and have removed the sentence “transfecting the cells first and infecting them after that “might be more similar to the latency model of viruses” and “We need to have a test that the TALEN plasmids transfection was prior to the MCMV infection” in our manuscript.
We have combined the original section 3.5, 3.5.1 and 3.5.2 to a new section “3.5. The effects of TALEN plasmids on MCMV titer” (P.11, Line 8-P.13, Line 4). Additionally, we have combined the original Figure 5, 7 to a new Figure 4 (P.12, Line 1-6) and the original Figure 6, 8 to Figure 6, 8 to a new Figure 5 (P.13, Line 1-4).”
17. If the TALEN pair MCMV 3-4 is more efficient in targeting/cleaving M80/80.5 (Figure 10), why does this not result in more efficient inhibition of viral replication (Figure 5, Figure 7)?
Response:
We are sorry to have an overstatement. By the results available in cell culture, we only can conclude MCMV3-4 is the most specific one and it is not sufficient to claim it is more efficient than the others. We have revised the statement all over the manuscript. The specificity is critical to avoid damaging normal and/or unrelated cells in animals; thus, we selected this one for our animal studies (P.15, Line 10-18).
18. Discuss potential explanations for different efficacy of TALENs between transfection pre- or post-infection?
Response:
We have added a paragraph in the “Discussion” section to discuss potential explanations for different efficacy of TALENs between transfection pre- or post-infection (P.19, Line 4-11).
19. Discuss innate immune (foreign DNA) response triggered by plasmid injection that may account for non-specific antiviral effects of TALENs.
Response:
We have added a paragraph in the “Discussion” section to discuss innate immune (foreign DNA) response triggered by plasmid injection that may account for non-specific antiviral effects of TALENs (P.19, Line 39-P.20, Line 8). Additionally, 6 related references (Reference 31-36, P.22, Line 18-29) have been added to support my statement.
20. Remove numbers from section/subsection headings.
Response:
Yes, we have removed them.

Reviewer 2 Report
This paper addresses the potential of eradicating latent CMV by performing proof of concept studies in mice.
The paper presents a lot of controls as separate figures which could be amalgamated to give a better flow.plus much is redundant. Is a growth curve of mcmv in 3T3 cells required?
However, my understanding of the data is that in the lytic data is that there are no major affects on viral replication - although no stats provided.
The potential impact I see is the reactivation data which suggests that TALENs reduce viraemia post reactivation. However, the major issue is that the KSHV TALENs have an effect as far as I can see. This seems to be a major issue moving forward as it argues that the mechanism is not specific?
Do the authors know if the latent virus is reduced prior to reactivation? If they aren't then the TALENs are targeting the reactivsting virus not the latent pool. This seems key for any attempt to use this strategy to eradicate a latent virus.
Author Response
Responses to the review report
Reviewer 2:
This paper addresses the potential of eradicating latent CMV by performing proof of concept studies in mice.
Response:
We thank the reviewer’s insightful comment for our manuscript. Attachment is our revised manuscript.
The paper presents a lot of controls as separate figures which could be amalgamated to give a better flow.plus much is redundant. Is a growth curve of mcmv in 3T3 cells required?
Response:
We have removed many reductant figures (e.g., original Figure 3, 12, 13, 14) and tables (e.g., Table 2) combined many figures (e.g., Figure 5, 6, 7, 8). Additionally, we rearranged the content of this manuscript to make it more logical and organized.
However, my understanding of the data is that in the lytic data is that there are no major effects on viral replication - although no stats provided.
Response:
For the virus titration assay in cell culture, the lytic data demonstrated significant inhibition on viral replication if plasmid transfection is prior to viral infection. When specific TALEN plasmids (MCMV1-2, 3-4 and 5-6) transfection was prior to the viral infection, we found the viral titers decreased about 65%, 75%, 25%, respectively, compared with the controls, using lipofectamine as a transfection reagent (Figure 4A, B, C). Additionally, the viral titers decreased about 50%, 60%, 25%, respectively, compared with the controls, using NKS11 as a transfection reagent (Figure 4D, E, F) (P.11, Line 11-P.12, Line 10). All the values were the average of triplicate experiments and indicated as Mean ± SD (standard deviation).
The potential impact I see is the reactivation data which suggests that TALENs reduce viremia post reactivation. However, the major issue is that the KSHV TALENs have an effect as far as I can see. This seems to be a major issue moving forward as it argues that the mechanism is not specific?
Response:
We have added a paragraph in the “Discussion” section to discuss innate immune (foreign DNA) response triggered by plasmid injection that may account for non-specific antiviral effects of TALENs (P.19, Line 39-P.20, Line 8). Additionally, 6 related references (Reference 31-36, P.22, Line 18-29) have been added to support my statement.
Do the authors know if the latent virus is reduced prior to reactivation? If they aren't then the TALENs are targeting the reactivating virus not the latent pool. This seems key for any attempt to use this strategy to eradicate a latent virus.
Response:
We are sorry that we do not have data to show if the latent virus is reduced prior to reactivation after TALENs treatment.
We have added a paragraph in the “Discuss” section to discuss “If they aren't then the TALENs are targeting the reactivating virus not the latent pool.” (P.19, Line 28-38). By the data shown in Table 4, we could demonstrate the viral load significantly increased after reactivation for all five organs of mice if there was no treatment (Group 1), compare with the treatment groups (Group 2-5) (P.17, Line 26-P.18, Line 22). In addition, we can confirm that it is impossible for the specific treatment group (Group 2) to be all ND (not detected) in the ie-1 DNA copy number assay for all five organs of mice, if the TALENs are only targeting the reactivating viruses not the latent pools (Table 4, P.18, Line 6-22). Thus, we consider that it is possible to remove latent viruses using this strategy.

Round 2
Reviewer 1 Report
The authors have addressed most of my comments. Although the presentation has been improved overall, the authors need to give the manuscript a few more rounds of proofreading to remove grammatical and other language errors, involving a native English speaker (if possible).
Author Response
We greatly appreciate the reviewer's help and comment. We have already given a few more rounds of proofreading to remove grammatical errors, cumbersome words and repetitive sentences all over the manuscript. Additionally, we revised the results "3.5" (P.11, Line 8-P.13, Line 4) and "3.6" (P.15, Line 11-20).
Reviewer 2 Report
The authors have made a substantial attempt to revise the paper. The inclusion of more tempered interpretations of the data are important. It is important to acknowledge that there are no data that categorically show that the latent virus has been eradicated and that this is an IFN effect.
Author Response
We thank the reviewer's insightful comment. We have added a sentence "However, there are no data that categorically show that the latent virus has been eradicated and that this is an IFN effect in our studies" in the discussion section (P. 20, Line 4-6).